# Evaluation of a Generative Adversarial Network to Improve Image Quality and Reduce Radiation-Dose during Digital Breast Tomosynthesis

**DOI:** 10.3390/diagnostics12020495

**Published:** 2022-02-14

**Authors:** Tsutomu Gomi, Yukie Kijima, Takayuki Kobayashi, Yukio Koibuchi

**Affiliations:** 1School of Allied Health Sciences, Kitasato University, Sagamihara 252-0373, Kanagawa, Japan; 2Department of Radiology, National Hospital Organization Takasaki General Medical Center, Takasaki 370-0829, Gunma, Japan; kijima.yukie.an@mail.hosp.go.jp; 3Department of Radiology, Kitasato University Kitasato Institute Hospital, Shirokane, Minato-ku, Tokyo 108-8642, Japan; kobataka@insti.kitasato-u.ac.jp; 4Department of Breast and Endocrine Surgery, National Hospital Organization Takasaki General Medical Center, Takasaki 370-0829, Gunma, Japan; koibuchi.yukio.ks@mail.hosp.go.jp

**Keywords:** digital breast tomosynthesis, generative adversarial networks, radiation-dose reduction, improve image quality

## Abstract

In this study, we evaluated the improvement of image quality in digital breast tomosynthesis under low-radiation dose conditions of pre-reconstruction processing using conditional generative adversarial networks [cGAN (pix2pix)]. Pix2pix pre-reconstruction processing with filtered back projection (FBP) was compared with and without multiscale bilateral filtering (MSBF) during pre-reconstruction processing. Noise reduction and preserve contrast rates were compared using full width at half-maximum (FWHM), contrast-to-noise ratio (CNR), peak signal-to-noise ratio (PSNR), and structural similarity (SSIM) in the in-focus plane using a BR3D phantom at various radiation doses [reference-dose (automatic exposure control reference dose: AECrd), 50% and 75% reduction of AECrd] and phantom thicknesses (40 mm, 50 mm, and 60 mm). The overall performance of pix2pix pre-reconstruction processing was effective in terms of FWHM, PSNR, and SSIM. At ~50% radiation-dose reduction, FWHM yielded good results independently of the microcalcification size used in the BR3D phantom, and good noise reduction and preserved contrast. PSNR results showed that pix2pix pre-reconstruction processing represented the minimum in the error with reference FBP images at an approximately 50% reduction in radiation-dose. SSIM analysis indicated that pix2pix pre-reconstruction processing yielded superior similarity when compared with and without MSBF pre-reconstruction processing at ~50% radiation-dose reduction, with features most similar to the reference FBP images. Thus, pix2pix pre-reconstruction processing is promising for reducing noise with preserve contrast and radiation-dose reduction in clinical practice.

## 1. Introduction

Digital tomosynthesis provides limited three-dimensional (3D) structural information about body structures by combining the advantages of digital imaging [1,2] and computed tomography. More specifically, digital breast tomosynthesis (DBT) reconstructs an entire image volume from a sequence of projection-view mammograms acquired within a small number of projection angles over a limited angular range to yield limited 3D structural information. Effects from the superposition of tissues are reduced with DBT, but in many situations, such as in dense breasts, such effects can persist. DBT decreases the camouflaging effects of the overlapping fibroglandular breast tissues, improves the conspicuity of subtle lesions, and could thus be used to improve the early detection of breast cancer [1,3,4].

To date, several digital mammography-based DBT systems have been developed [5,6,7], and there are ongoing studies aiming to define its utility and improvements [1,8]. Wu et al. evaluated image quality using a conventional reconstruction algorithm (filtered back projection; FBP [9]), statistical iterative reconstruction (IR) algorithms (maximum likelihood expectation maximization; MLEM [3]), and simultaneous IR algorithms (the simultaneous IR technique; SIRT [10]) and concluded that the MLEM algorithm provided a good balance between low- and high-frequency features [3], and the exploration of various DBT reconstruction methods have been reported [11,12,13,14]. Other researchers used a total variation–minimization algorithm (adaptive steepest descent projection onto convex sets) [15] with a gradient-based penalty term to enhance microcalcifications (MCs) on DBT images [16]. On the other hand, another study quantitatively compared DBT algorithms in terms of image quality and radiation doses [17]. In that report, IR was found to effectively decrease quantum noise and radiation exposure; however, the evaluation in that study was limited and merely compared existing methods (FBP vs. IR: SIRT and MLEM).

With the aim of overcoming the drawbacks of previous algorithms, a recent report described the development of an improved processing method for iterative DBT reconstruction (multiscale bilateral filtering; MSBF) [18] with the simultaneous algebraic reconstruction technique algorithm [10]. Specifically, this method aimed to improve the contrast of MCs without compromising the image qualities of masses and soft-tissue background structures. The previous study evaluated only MCs and not masses.

Existing DBT techniques used in clinical diagnostic studies have enabled the visualization of fine tissue structures with a shorter scan time. Nevertheless, all DBT systems are limited by the issue of patient radiation exposure, which highlights the need to preserve contrast in order to improve the sharpness of the image and detectability of the object. Furthermore, radiographic images can be degraded by quantum mottle, a consequence of spatial incident photon fluctuation. Quantum mottle is inversely associated with exposure, and therefore, any decrease in patient dose would be restricted by the degree of quantum mottle, even with a perfect detector. Even with a perfect detector, other factors are present, including X-ray scatter and X-ray spectrum, number of views, and other factors, to avoid minimizing image reconstruction and image processing, both of which are very important. Therefore, further decreases in patient doses and improvements in detection rely on innovations such as a new detector type, alternative X-ray sources, and an algorithm that improves image quality by incorporating suitable approaches.

Moreover, DBT involves the reconstruction of images limited by a low signal-to-noise ratio due to the superposition of several low-exposure projection images. Furthermore, this characteristic causes a concurrent loss of plane-relevant details, which reduces the contrast of the reconstructed images. Several methods have been proposed to suppress this irrelevant plane information and enhance the image quality of DBT [18,19]. In reconstructed DBT images, noise further affects the visibility and detectability of subtle MCs. To overcome this limitation, several noise-suppression techniques have been proposed to enhance MCs [16,20,21]. However, most of the existing regularization methods for DBT reconstruction were designed for general image applications and are driven by local gradients [22,23].

In the accelerating evolution of deep learning, the transition from a convolutional neural network [24] to a generative adversarial network (GAN) [25,26] has contributed to digital tomosynthesis imaging [27,28,29,30,31,32,33,34,35,36]. Prior studies reported that GANs are particularly useful for reducing metal artifacts [29,30] and noise [27] and are expected to contribute to improvements in image quality processes to reduce the exposure dose. Some studies have recently reported the usefulness of deep learning to improve image quality and reduce noise in tomosynthesis [27,29]. Noise and radiation-dose reductions using deep learning for digital tomosynthesis of the breast and metal artifact reduction are possible [27,30]. Thus, application of deep learning can be used to improve image quality further and reduce the radiation-dose. In the DBT imaging field, recent reports have detailed the detection of masses and the image quality improvement process that introduces deep learning [28,31,32,33,34,35]. With regard to image quality improvement processing that uses conditional GAN (cGAN, or pix2pix) [25], “pix2pix,” which approximates the object image to the referenced image using the concept of an adversarial network using “generator” and “discriminator,” has been shown to be useful for noise reduction. cGAN provides a powerful image translation framework that works well in many areas. In addition to cGAN, cycleGAN can be considered, but it requires at least two discriminators and generators, which complicates the structure. Therefore, cGAN can be used as a general-purpose solution to the image-to-image translation problem. Using the conventional approach to image processing and image reconstruction, it is difficult to accelerate the detection of masses and preserve the normal structure with accuracy [11,12,13,14,15,16,18,37]. In particular, as the noise associated with low dose imaging is increased, there is a tradeoff in the acceleration of the detection of masses and preservation of normal structure (e.g., structural distortion, oversmoothing or sharpness, occurrence artifact, etc.). However, with the use of pix2pix, it might be possible to overcome the problems of the conventional method in DBT imaging.

Studies conducted to date have quantitatively compared various DBT algorithms in terms of image quality and radiation doses [38,39]. Although those reports demonstrated that IR could effectively decrease quantum noise and radiation exposure, the evaluations were limited and merely compared existing methods. In a related recent report, Gao et al. reported that denoising a deep convolutional neural network using adversarial training was useful for improving the MCs contrast in DBT using in silico data and applied to physical phantom images as a learning set [27]. Among the studies using deep learning, there are no reports on the quantitative evaluation of improvements in image quality or dose reduction of MCs and the detection of masses under various conditions with automatic exposure control (AEC) as the referenced dose because of changes in breast thickness. In particular, considering that pix2pix has the potential to reduce the dose and improve image quality, it can be expected that this logic (image-to-image translation process, in which a low dose image can be applied to the reference dose image) can be applied to DBT to improve the acquisition of image quality deterioration under low dose.

In this paper, we report our experience using the application of pix2pix pre-reconstruction processing (FBP reconstruction after preprocessing pix2pix) to improve image quality with dose reduction and amend processing. Because the usefulness of preprocessing has been reported in improving image quality using deep learning processing of tomosynthesis [27], in this study we performed deep learning processing (pix2pix) on projection-based data. In addition, because the purpose of this study was not to compare the reconstruction algorithms, we used the exact solution (FBP) for evaluation. Our proposed preprocessing pix2pix exploits both the improved detection of MCs and the preservation of normal structures to improve both spatial resolution and contrast preservation. Our proposed pix2pix pre-reconstruction processing may overcome a previously unresolved problem associated with conventional algorithms, namely, the improved detection of MCs and preserved contrast of masses, by correcting reconstruction processing with dose reduction. In addition, we evaluate the usefulness of pix2pix pre-reconstruction processing for the purpose of improving image quality under dose reduction. Specifically, pix2pix pre-reconstruction processing is applied to the projection data (reference dose [automatic exposure control reference dose: AECrd] and low dose [approximately 50% and 75% reduction of AECrd]) when the phantom thickness is changed and the reconstructed image (FBP) with physical evaluation (spatial resolution, contrast, error, similarity).

## 2. Materials and Methods

### 2.1. DBT

This study used a DBT system (Selenia Dimensions; Hologic Inc., Bedford, MA, USA) that consists of an X-ray tube with a 0.3 mm focal spot (tube target: W, filtration: 0.7 mm aluminum equivalent) and a digital flat-panel amorphous selenium detector. A total acquisition time of 3.7 s and an acquisition angle of 15° were set for all DBT procedures. The projection images were sampled during a single tomographic pass (15 projections, 1280 × 2048 matrix). To produce reconstructed tomograms of the required height, we used a 512 × 1024 matrix with 32 bits (single-precision floating number) per image.

### 2.2. Phantom Specifications

The BR3D phantom (model 020; Computerized Imaging Reference Systems, Inc., Norfolk, VA, USA) comprises multiple heterogeneous slabs and is intended to mimic the composition of glandular and adipose tissues and parenchymal patterns in the human breast. The slabs are composed of epoxy resins with X-ray attenuation properties corresponding to 50% glandular or 50% adipose breast tissue. In the phantom, the target slab was surrounded by nontarget slabs (top: 30, 40, 50 mm; bottom: 10 mm).

### 2.3. Radiation-Dose Measurement

For each radiation-dose setup, the reference radiation-dose (AECrd = exposure condition at 40-, 50-, and 60-mm thickness and predetermined tube voltage, current, and exposure time) was set at 29 kVp 145 mA 310 ms, 31 kVp 170 mA 310 ms, and 33 kVp 200 mA 320 ms. The average glandular dose (AGD) for DBT was calculated according to Dance et al. [40]. To measure the radiation exposure, we used a Piranha dosimeter (RTI Electronics AB, Mölndal, Sweden) to convert the established exposure condition into the AGD. The AGD results were as follows: reference radiation-dose (AECrd), 1.36 mGy (40 mm), 1.77 mGy (50 mm), 2.35 mGy (60 mm); approximately 50% reduction of AECrd, 0.66 mGy (40 mm), 0.74 mGy (50 mm), 1.13 mGy (60 mm); and approximately 75% reduction of AECrd, 0.31 mGy (40 mm), 0.37 mGy (50 mm), and 0.61 mGy (60 mm).

### 2.4. Pix2pix

Pix2pix is a GAN that trains generators and discriminators by providing various additional information and allowing it to be conditioned. Because of constraints on the additional information, the generator produces certain types of output, and the discriminator accepts only additional information that matches the real sample. The training objectives of the discriminator and generator can be expressed mathematically as follows:(1)$$\begin{array}{ll} \underset{D}{\min}L_{GANc}= & E_{p_{ld,}p_{ref}}[\log D\left(p_{ld,}p_{ref}\right)]\\ & \;+E_{p_{ld,z}}[\log(1-D(p_{ld,}G\left(p_{ld,}z\right))] \end{array}$$
(2)$$\underset{G}{\min}L_{GAN_{l1}}=E_{p_{ld,}p_{ref,z}}\left[{\left\|P_{ref}-G\left(p_{ld,}z\right)\right\|}_{1}\right]$$
where $P_{ld}$ is the low dose projection domain, $P_{ref}$ is the reference dose projection domain, *D* is the discriminator, *G* is the generator, and *z* is the random noise vector (Gaussian noise). The training data set included 180 projection images, and each of the corresponding images related to the input image pair ($P_{ld}$(90), $P_{ref}$(90)) were randomly selected as the training set.

The pix2pix was developed to solve the following problem:(3)$$p_{ld}{}^{†}=\arg\underset{G}{\min}\underset{D}{\max}L_{GANc}\left(G,D\right)+\alpha \underset{G}{\min}L_{GAN_{l1}}(G).$$
where α controls the relative importance of the two objectives $L_{GANc}$ and $L_{GAN_{l1}}$. In this study, α was set to 50, the initial learning rate was set to 0.0002, and the momentum parameters were set to $\beta_{1}=0.5{,}_{}^{}\beta_{2}=0.999$ [25].

In pix2pix, we used the Adam optimization algorithm [41] with a batch size of 1. Appendix A (Table A1) shows the architecture of the building components.

### 2.5. Optimization Parameters for Epochs

The optimization epochs in the pix2pix network were evaluated based on the mean square error (MSE) [42] and structural similarity (SSIM) [43] for the projection image (straightforward on the detector; 0 degree). The MSE of the identified projection image can be obtained as follows:(4)$$MSE=\frac{1}{mn}{\sum_{i=0}^{m-1}\sum_{j=0}^{n-1}\left[I_{DBT\_ref}(i,j)-I_{DBT\_low}(i,j)\right]}^{2}$$
where $I_{DBT\_ref}(i,j)$ is the (*i*, *j*)th entry of the reference dose projection image, and $I_{DBT\_low}(i,j)$ is the (*i*, *j*)th entry of the low dose projection image in each epoch.

The SSIM index between pixel values *i* and *j* was calculated as follows:(5)$$SSIM(i,j)={\left[L_{umi}(i,j)\right]}^{\varepsilon }\cdot {\left[C_{ont}(i,j)\right]}^{\phi }\cdot {\left[S_{tru}(i,j)\right]}^{\eta }$$
where $L_{umi}$ is the luminance, $C_{ont}$ is the contrast, and $S_{tru}$ is the structure (ε=ϕ=η=1.0).

The mean SSIM (MSSIM) was then used to evaluate the overall image quality as follows:(6)$$MSSIM[I_{DBT\_ref}(i,j),I_{DBT\_low}(i,j)]=\frac{1}{Q}\sum_{r=1}^{Q}SSIM(i_{r},j_{r})$$
where $i_{r}$ and $j_{r}$ are the image contents at the *r*th pixel and *Q* is the number of pixels in the image.

Optimization was evaluated based on the MSE and MSSIM for 40-mm phantom thickness. The lowest MSE, highest MSSIM, and epochs were selected as the optimum parameters.

### 2.6. Evaluation of Image Quality

The DBT system-derived real projection data were used for FBP reconstruction. We used MATLAB (MathWorks, Natick, MA, USA) to reconstruct and process all images (custom script for MATLAB environment). For each phantom image, we calculated the full width at half-maximum (FWHM), contrast-to-noise ratio (CNR), peak signal-to-noise ratio (PSNR), and SSIM and in the in-focus plane to evaluate the effects of each processing method. The target images of FWHM, CNR, PSNR, and SSIM were evaluated by selecting different in-focus planes in the longitudinal direction. For the FWHM in the in-focus plane (0.196, 0.23, and 0.29 mm; CaCO_3_), the derived spatial resolution was evaluated as a quantitative measure of the reconstructed image quality. Subsequently, the FWHMs were measured for selected intensity profiles intersecting the five MCs on reconstructed DBT slices. Next, four summation neighboring (i.e., parallel and perpendicular to the X-ray sweep direction) were arranged to obtain an intensity profile.

In addition, the contrast was derived from the CNR in the in-focus plane (3.9 and 4.7 mm φ; spheroidal masses [epoxy resin]) and used to quantitatively measure the reconstructed image quality. Tomosynthesis applications frequently use the CNR to estimate low-contrast detectability. In this study, we defined the CNR as follows:(7)$$CNR=\frac{\mu_{Feature}-\mu_{BG}}{\sigma_{BG}}$$
where $\mu_{Feature}$ represents the mean object pixel value, $\mu_{BG}$ represents the mean background area pixel value, and $\sigma_{BG}$ represents the standard deviation of the background pixel values (set up in four locations around the masses; up, down, left, right). Of these parameters, the latter includes both photon statistics and electronic noise from the results, and structural noise that might obscure the object of interest. For all regions of interest (ROIs) used to measure the CNR, the sizes were adjusted to an internal signal (circular ROI, 3.9 mm [diameter: 28 pixels], 4.7 mm [diameter: 40 pixels]). To assess the improvement of image quality on each in-focus plane image, the conventional algorithms (FBP reconstruction from the MSBF processing projections) were compared.

PSNR represents the ratio of the maximum power that a signal can take and the noise that causes degradation, which affects the reproducibility of image quality on each in-focus plane image. The PSNR was defined as follows:(8)$$PSNR=10\log_{10}\left(\frac{PV^{2}}{MSE}\right)$$

We used a *PV* value of 1.0 because we assumed that the image data (single-precision floating number) was in the range [0, 1.0]. The *MSE* was calculated between the reference dose and low dose FBP images.

This study compared the performance of pix2pix pre-reconstruction processing with that of MSBF pre-reconstruction processing. Here, the parameter settings ($\sigma_{d}$) were determinants of the image quality. Except for $\sigma_{d}$, all other set values were as previously reported [18]. In this study, the parameter $\sigma_{d}$ was chosen to be 1.0 from the perspective of contrast preservation, in accordance with a previous study (*α*: 0.375, *w*_α_: 5 × 5 Laplacian filter, *σ_r_*: 0.01) [18].

The impulse shape of each reconstructed image was restored using two-dimensional image filtering, which multiplied the Fourier transform by a Ramachandran and Lakshminarayanan kernel, which generally produced precise 3D reconstruction images [11].

In this study, we compared the FWHM values with and without MSBF pre-reconstruction processing at different radiation doses between the four groups (reference dose, low dose without MSBF pre-reconstruction processing, low dose with pix2pix pre-reconstruction processing, and low dose with MSBF pre-reconstruction processing). The numbers of samples in the groups were reference dose (0.19 mm: 30; 0.23 mm: 36; and 0.29 mm: 36), low dose without MSBF pre-reconstruction processing (0.19 mm: 60; 0.23 mm: 72; and 0.29 mm: 72), low dose with pix2pix pre-reconstruction processing (0.19 mm: 60; 0.23 mm: 72; and 0.29 mm: 72), and low dose with MSBF pre-reconstruction processing (0.19 mm: 56; 0.23 mm: 70; and 0.29 mm: 72). Statistical analyses were performed using IBM SPSS Statistics version 21.0 for Windows (SPSS Inc., Chicago, IL, USA). Probability (*p*) values < 0.05 were considered statistically significant.

## 3. Results

### 3.1. Optimization Parameters

After measuring the MSE and SSIM of each training network at different phantom thicknesses (40, 50, and 60 mm) and radiation doses (approximately 50% and 75% reduction of AECrd), the optimal epoch was selected at the lowest MSE and highest SSIM. Using the results of the optimization verification, each training network image was generated by setting 300 epochs for pix2pix, and then evaluated and compared with those of the images obtained using the conventional approach with and without MSBF pre-reconstruction processing (Figure 1). The training was performed on a TITAN RTX (24 GB of memory) GPU. The total calculation time required to process pix2pix was 13.63 h (epochs 300).

### 3.2. Image Quality

Figure 2, Figure 3 and Figure 4 show the reconstructed images of the B3RD phantom acquired with pix2pix pre-reconstruction processing and each of the established methods for reconstruction with and without MSBF pre-reconstruction processing at a reduced radiation-dose (approximately 50% and 75% reduction of AECrd) and reference radiation-dose. Remarkably, the DBT images produced using pix2pix pre-reconstruction processing showed reduced noise and preservation of contrast in the radiographic vertical and horizontal direction, specifically in the peripheral regions of the MCs and masses. On the other hand, images produced with the help of MSBF pre-reconstruction processing demonstrated noise. Comparison of the differences between pix2pix pre-reconstruction processing and the conventional approach with and without MSBF pre-reconstruction processing resulted in the smallest with MSBF pre-reconstruction processing for noise reduction. With MSBF pre-reconstruction processing showed a certain reduction in noise, but the lack of preservation of contrast generated from around the MCs was remarkable. Without MSBF pre-reconstruction processing, higher noise levels increased with radiation-dose reduction, resulting in a deterioration in image quality.

Figure 5, Figure 6 and Figure 7 depict the areas of MCs in a BR3D phantom and a plot of the FWHM results. For MCs of 0.23 mm or greater, the reference dose and pix2pix pre-reconstruction processing (approximately 50% and 75% reduction of AECrd) showed equal average and median characteristics, but at 0.19 mm with approximately 75% reduction of AECrd, the result deteriorated at a horizontal direction of 50 mm or greater. At a pix2pix pre-reconstruction processing of up to approximately 50% reduction of the AECrd, the structure of the MCs was preserved regardless of the BR3D phantom thickness or MCs size, as compared with the reference dose. A comparison between reference dose and without MSBF pre-reconstruction processing (approximately 50% and 75% reduction of AECrd) showed comparable mean and median characteristics at greater than 0.23 mm; deterioration was observed at a vertical direction of all BR3D phantom thicknesses at a vertical direction of 0.19 mm. Comparisons between the reference and with MSBF pre-reconstruction processing (approximately 50% and 75% reduction of AECrd) deteriorated at all BR3D phantom thicknesses and MCs sizes. In particular, the result was affected with a size of 0.23 mm or greater. For all sizes of MCs, the differences in the FWHM, except for pix2pix pre-reconstruction processing compared with the reference and without MSBF pre-reconstruction processing (approximately 50% and 75% reduction of AECrd), were not statistically significant (Table 1, Table 2 and Table 3). For MCs of all sizes, the differences in the FWHM, except for pix2pix pre-reconstruction processing compared with MSBF pre-reconstruction processing (approximately 50% and 75% reduction of AECrd), were statistically significant (*p* < 0.05; Table 1, Table 2 and Table 3). These FWHM results showed that pix2pix pre-reconstruction processing was preserved in areas with MCs of BR3D phantom.

Figure 8 and Figure 9 show the whole image areas of the BR3D phantom and a plot of the SSIM and PSNR results. With regard to the similarity and error of the reference dose, pix2pix pre-reconstruction processing showed high similarity and low error under all conditions, regardless of low dose level (approximately 50% and 75% reduction of AECrd) and BR3D phantom thickness. Regarding PSNR, in pix2pix pre-reconstruction processing, PSNR decreased and errors increased in parts of in-focus planes (Figure 9a) for approximately 75% reduction of AECrd. MSBF pre-reconstruction processing showed high similarity compared without pre-reconstruction processing, but the result was that the error was large.

Figure 10 depicts the placement of the ROI over an image of the BR3D phantom and a plot of the CNR results. With regard to the contrast of masses, with MSBF pre-reconstruction processing was the highest, followed by pix2pix pre-reconstruction processing, and without MSBF pre-reconstruction processing showed the lowest contrast characteristics for 4.7 mm mass. For pix2pix, the CNR was equivalent to that of reference under dose reduction (approximately 50% and 75% reduction of AECrd).

There was no deterioration in image quality with pix2pix pre-reconstruction processing under the low dose level (approximately 50% reduction of AECrd) conditions associated with changes in BR3D phantom thickness in FWHM, SSIM, and MSE, except for the contrast of masses. This result indicates that pix2pix may be useful for radiation-dose-reduction technology without a subsequent deterioration in image quality.

## 4. Discussion

Our experimental results clearly demonstrated the ability of pix2pix pre-reconstruction processing to improve the quality of DBT images for the low dose condition. In this study, the in-focus plane intensities of pix2pix pre-reconstruction processing, as compared with existing MSBF pre-reconstruction processing, improved spatial resolution, similarity, and image error without deterioration of the MCs images with whole image. Furthermore, pix2pix pre-reconstruction processing has the potential to reduce the radiation-dose by approximately 50% reduction of AECrd. Thus, pix2pix pre-reconstruction processing is a promising new option for imaging denoising, as it generated noise-reduced images and reduced radiation doses that were far superior to those obtained from images processed using conventional algorithms. The flexibility of pix2pix pre-reconstruction processing in the choice of imaging parameters, which is based on the desired final images and low dose DBT imaging conditions, promises increased usability.

The projection-space combination of adversarial training approaches described here can be used to generate images to formulate denoising as a deep learning algorithm for projection completion problems, with the aim of improving the generalization and robustness of the framework. Because the direct regression of accurate projection data is difficult [17,18], we propose incorporating the prior projection image generation procedure and adopting a combination of adversarial networks and a projection completion strategy. This method can improve image quality by reducing noise while preserving masses and normal structures, which are common drawbacks of projection completion-based adversarial training methods. Therefore, we believe that our adversarial training approaches could effectively reduce noise in actual practice.

The ability of pix2pix pre-reconstruction processing to obtain denoising and contrast-preserving images and to reduce the radiation-dose by approximately 50% reduction of AECrd (Figure 5, Figure 6, Figure 7, Figure 8 and Figure 9) may be due to the benefits of the first process, pix2pix. The image-to-image translation framework of pix2pix requires fully associated images. pix2pix is different from conventional noise reduction by reconstruction/processing, because it can preserve the structures and reduce the noise, therefore solving this problem. pix2pix has a generator that attempts to minimize this objective against an adversarial discriminator that tries to maximize. The generators use a U-Net [44] structure, and the two discriminators have a Patch-GAN-based structure [45] for learning. By applying another style to the image during the translation process, the low dose projection image can then be applied to the reference dose projection image.

In MSBF pre-reconstruction processing, Laplacian pyramid decomposition (LPD) is used to achieve multiscale structure decomposition during MSBF pre-reconstruction processing. However, this function is not unique to LPD, and other multiscale analysis methods may be sufficient. In this regard, however, the MCs detected via MSBF pre-reconstruction processing may not have strong directional geometric features. Therefore, a directional multiscale analysis method, such as wavelet transform, may not be superior to the LPD method [18].

The image artifacts from MCs leads to the appearance of noticeable objects comprising artifact-free voxels that contrast with the background. These artifacts from MCs are a drawback of the FBP algorithm and are conspicuous when images generated using this method are compared with artifact-free images. Therefore, based on the results of this study, future research should consider conducting evaluations using the IR algorithm.

There were some limitations to this phantom study. First, the materials constituting the BR3D phantom were only simulations of the mammary gland, because actual mammary gland tissues were not tested. Alternatively, we suppose that the consistency of the BR3D phantom indicates that it is an accurate representation of actual mammary gland tissue. Second, we did not perform a clinical study using human subjects. The utility of pix2pix pre-reconstruction processing was confirmed by basic evaluation. In a future observational study, we plan to investigate the correlation between spatial resolution and contrast. We believe that pix2pix pre-reconstruction processing will allow for optimization of the use of dose in future DBT imaging and radiation-dose reduction technology and improve the accuracy of medical images. Third, the experiment was evaluated by a single vendor system. We think that a study using a multi-vendor system is necessary. Fourth, optimization of projection data analysis of MSE and SSIM was for the central projection only; the performance at other angles will depend on the object’s shape. Evaluations will be relevant to checking the sensitivity of optimization according to the projection view angle with respect to the detector. Fifth, evaluation of the in-focus slice of the reconstruction volume only; consideration of any out-of-plane features/artefacts is necessary. Sixth, use of the FBP algorithm only, without optimization (kernel; Ramachandran and Lakshminarayanan). This leaves opportunities for future work (i.e., evaluation of IR algorithms). Seventh, we believe that the CNR evaluation of masses is limited and requires assessment using a wider variety of sizes, shapes and margin types (e.g., smooth, spiculated) to improve accuracy. In addition, we considered using more advanced methods such as a detectability index [27], where the influence of anatomical noise can be included.

## 5. Conclusions

This phantom study revealed that an approximately 50% reduction in radiation-dose is feasible using our proposed pix2pix pre-reconstruction processing. The pix2pix pre-reconstruction processing was particularly useful in reducing noise and yielded better results equivalent to that of the reference dose, with no significant difference in the statistical results (0.196 mm: *p* = 0.996; 0.23 mm: *p* = 0.886; 0.29 mm: *p* = 0.321) in terms of preserving the structure of MCs as compared with variant phantom thickness. Thus, pix2pix shows promise for integration into the clinical application workflow to reduce image noise while maintaining image quality in breast tomosynthesis.

## Figures and Tables

**Figure 1 diagnostics-12-00495-f001:**
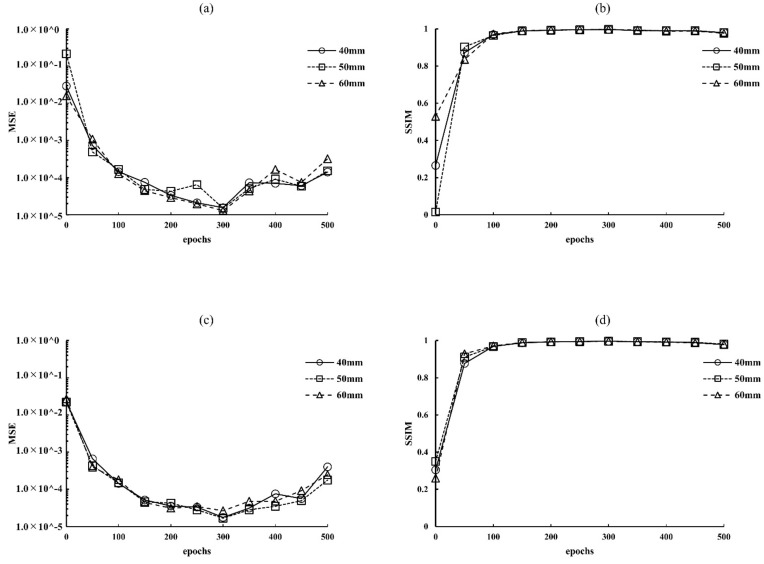
Optimization results for parameter (epochs) determination for pix2pix pre-reconstruction processing at different radiation-dose levels: (**a**) 0.66 mGy (approximately 50% of automatic exposure control reference dose [AECrd]), (**b**) 0.31 mGy (approximately 50% reduction of AECrd), (**c**) 0.66 mGy (approximately 75% reduction of AECrd), and (**d**) 0.31 mGy (approximately 75% reduction of AECrd). BR3D phantom thickness: 40 mm.

**Figure 2 diagnostics-12-00495-f002:**
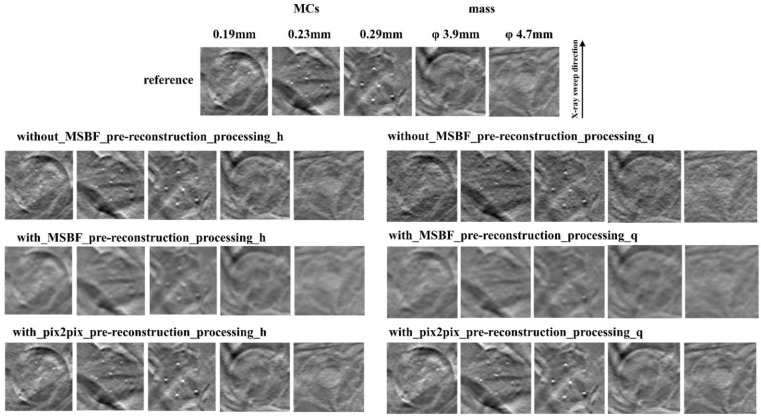
Comparisons between pix2pix pre-reconstruction processing and conventional multiscale bilateral filtering (MSBF) pre-reconstruction processing with and without noise reduction (microcalcifications (MCs) [showing window: 0.41–0.70]; masses [3.9 mm: 0.41–0.70, 4.7 mm: 0.35–0.64]) in the in-focus plane (BR3D phantom thickness: 40 mm). The display referring to the image contrast of the BR3D phantom was changed for visual comparison of the signal and background gray levels. h: approximately 50% reduction of automatic exposure control reference dose (AECrd), q: approximately 75% reduction of AECrd.

**Figure 3 diagnostics-12-00495-f003:**
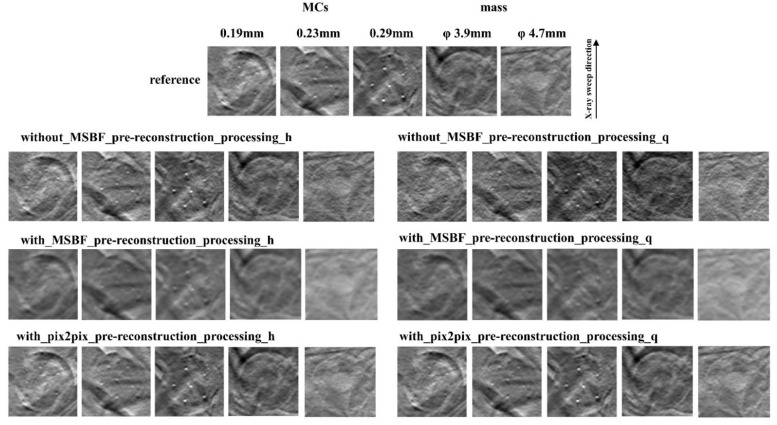
Comparisons between pix2pix pre-reconstruction processing and conventional multiscale bilateral filtering (MSBF) pre-reconstruction processing with and without noise reduction (microcalcifications (MCs) [showing window: 0.39–0.68]; masses [3.9 mm: 0.39–0.68; 4.7 mm: 0.28–0.57]) in the in-focus plane (BR3D phantom thickness: 50 mm). The display referring to the image contrast of the BR3D phantom was changed for visual comparison of the signal and background gray levels. h: approximately 50% reduction of automatic exposure control reference dose (AECrd); q: approximately 75% reduction of AECrd.

**Figure 4 diagnostics-12-00495-f004:**
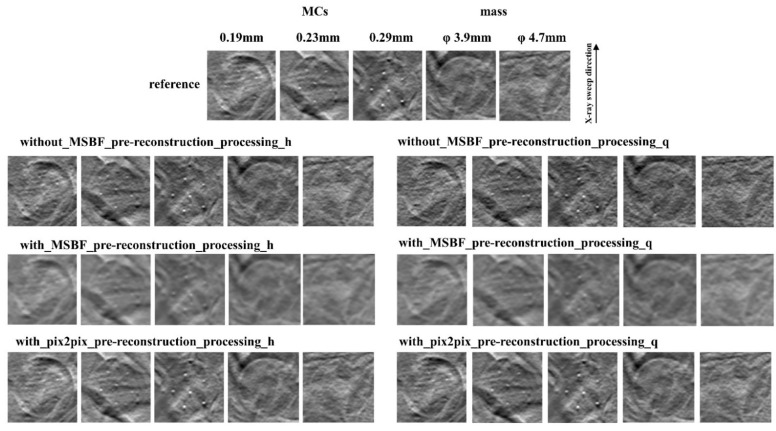
Comparisons between pix2pix pre-reconstruction processing and the conventional multiscale bilateral filtering (MSBF) pre-reconstruction processing with and without noise reduction (microcalcifications (MCs) [showing window: 0.30–0.59]; masses [3.9 mm: 0.30–0.59; 4.7 mm: 0.23–0.52]) in the in-focus plane (BR3D phantom thickness; 60 mm). The display referring to the image contrast of the BR3D phantom was changed for visual comparison of the signal and background gray levels. h: approximately 50% reduction of automatic exposure control reference dose (AECrd); q: approximately 75% reduction of AECrd.

**Figure 5 diagnostics-12-00495-f005:**
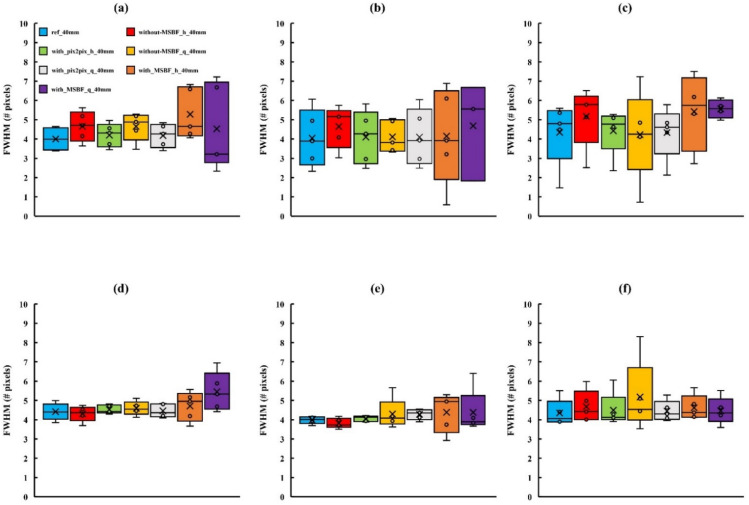
Comparisons of the full width at half-maximum (FWHM) determined for in-focus plane images obtained using pix2pix pre-reconstruction processing and reference, with and without multiscale bilateral filtering (MSBF) pre-reconstruction processing, for low dose with variant phantom thickness [microcalcifications (MCs) size: 0.19 mm]. (**a**) Vertical direction 40 mm; (**b**) vertical direction 50 mm; (**c**) vertical direction 60 mm; (**d**) horizontal direction 40 mm; (**e**) horizontal direction 50 mm; and (**f**) horizontal direction 60 mm. h: approximately 50% reduction of automatic exposure control reference dose (AECrd); q: approximately 75% reduction of AECrd.

**Figure 6 diagnostics-12-00495-f006:**
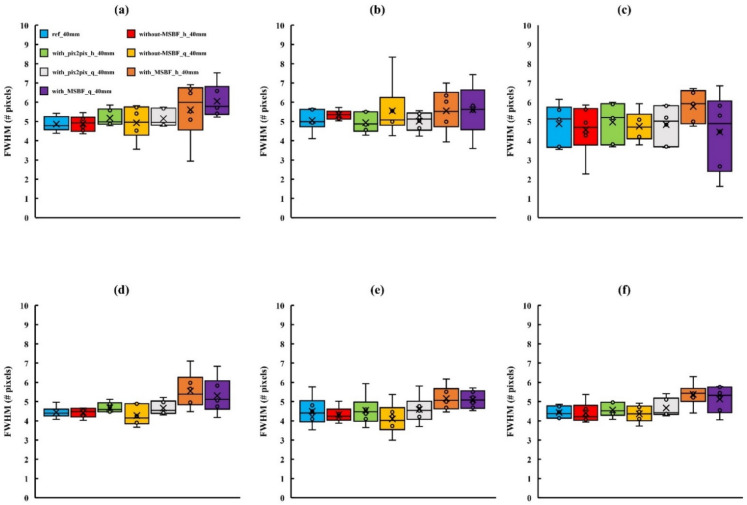
Comparisons of the full width at half-maximum (FWHM) determined for in-focus plane images obtained using pix2pix pre-reconstruction processing and reference, with and without multiscale bilateral filtering (MSBF) pre-reconstruction processing, for low dose with variant phantom thickness [microcalcifications (MCs) size: 0.23 mm]. (**a**) Vertical direction 40 mm; (**b**) vertical direction 50 mm; (**c**) vertical direction 60 mm; (**d**) horizontal direction 40 mm; (**e**) horizontal direction 50 mm; and (**f**) horizontal direction 60 mm. h: approximately 50% reduction of automatic exposure control reference dose (AECrd); q: approximately 75% reduction of AECrd.

**Figure 7 diagnostics-12-00495-f007:**
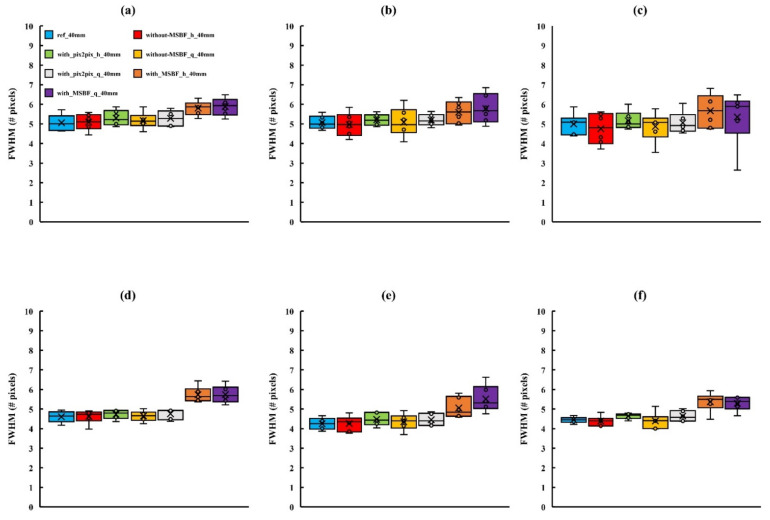
Comparisons of the full width at half-maximum (FWHM) determined for in-focus plane images obtained using pix2pix pre-reconstruction processing and reference, with and without multiscale bilateral filtering (MSBF) pre-reconstruction processing, for low dose with variant phantom thickness [microcalcifications (MCs) size: 0.29 mm]. (**a**) Vertical direction 40 mm; (**b**) vertical direction 50 mm; (**c**) vertical direction 60 mm; (**d**) horizontal direction 40 mm; (**e**) horizontal direction 50 mm; and (**f**) horizontal direction 60 mm. h: approximately 50% reduction of automatic exposure control reference dose (AECrd); q: approximately 75% reduction of AECrd.

**Figure 8 diagnostics-12-00495-f008:**
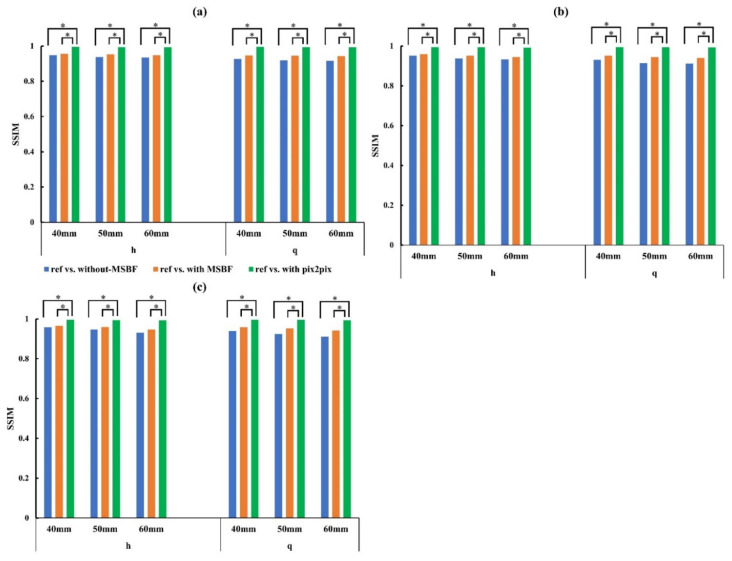
Plots of the structural similarity (SSIM) vs. reference vs. without pre-reconstruction processing, reference vs. with multiscale bilateral filtering (MSBF) pre-reconstruction processing, reference vs. pix2pix pre-reconstruction processing from the in-focus plane for low dose with variant phantom thickness. (**a**) In-focus plane for evaluation of 0.19- and 0.23-mm microcalcifications (MCs); (**b**) in-focus plane for evaluation of 0.29-mm MCs and 3.9-mm masses; (**c**) in-focus plane for evaluation of 4.7-mm mass. h: approximately 50% reduction of automatic exposure control reference dose (AECrd); q: approximately 75% reduction of AECrd. (Tukey–Kramer test; *p* < 0.05 indicates a significant difference, *: significant).

**Figure 9 diagnostics-12-00495-f009:**
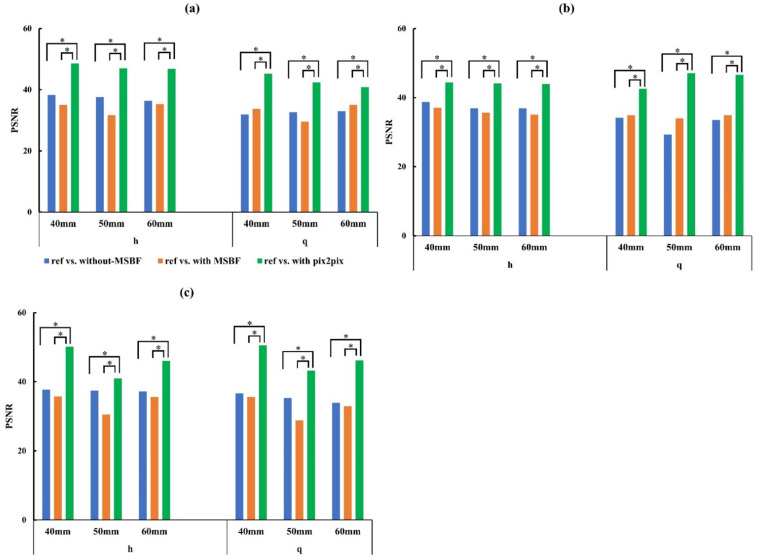
Plots of the peak signal-to-noise ratio (PSNR) vs. reference vs. without pre-reconstruction processing, reference vs. with multiscale bilateral filtering (MSBF) pre-reconstruction processing, reference vs. pix2pix pre-reconstruction processing from the in-focus plane for low dose with variant phantom thickness. (**a**) In-focus plane for evaluation of 0.19- and 0.23-mm microcalcifications (MCs). (**b**) In-focus plane for evaluation of 0.29-mm MCs and 3.9-mm masses. (**c**) In-focus plane for evaluation of 4.7-mm mass. h: approximately 50% reduction of automatic exposure control reference dose (AECrd); q: approximately 75% reduction of AECrd. (Tukey–Kramer test; *p* < 0.05 indicates a significant difference, *: significant).

**Figure 10 diagnostics-12-00495-f010:**
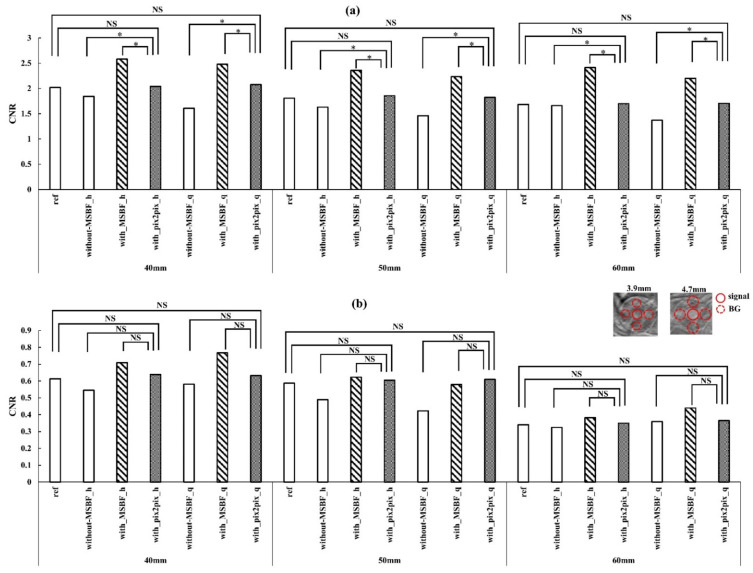
Plots of the contrast-to-noise ratio (CNR) vs. pre-reconstruction processing, with and without multiscale bilateral filtering (MSBF) pre-reconstruction processing from the in-focus plane. Comparisons of the CNR of the in-focus plane images obtained via the reference, low dose [approximately 50% and 25% of automatic exposure control reference dose (AECrd)] with and without pre-reconstruction processing with varying phantom thicknesses. (**a**) 4.7 mm mass; (**b**) 3.9 mm mass. The in-focus plane image shows the masses and background areas of the CNR. h: approximately 50% reduction of AECrd; q: approximately 75% reduction of AECrd. (Tukey–Kramer test; *p* < 0.05 indicates a significant difference, *: significant, NS: not significant).

**Table 1 diagnostics-12-00495-t001:** Spatial resolution of microcalcifications (MCs) performances of tomosynthesis processing methods. (Tukey–Kramer test; *p* < 0.05 indicates a significant difference).

MCs: 0.196 mm					
Variable	Difference	Standard Error	*p*	95% CI *	
				Lower Limit	Upper Limit
Ref vs. Without pre-reconstruction processing	−0.3407	0.24671	0.513	−0.9799	0.2895
vs. MSBF	−0.6529	0.24963	0.047	−1.2996	−0.0061
vs. pix2pix	−0.0553	0.24671	0.996	−0.6945	0.5838
Without pre-reconstruction processing vs. Ref	0.3407	0.24671	0.513	−0.2985	0.9799
vs. MSBF	−0.3122	0.20500	0.426	−0.8433	0.2190
vs. pix2pix	0.2854	0.20143	0.490	−0.2365	0.8073
MSBF vs. Ref	0.6529	0.24963	0.047	0.0061	1.2996
vs. Without pre-reconstruction processing	0.3122	0.20500	0.426	−0.2190	0.8433
vs. pix2pix	0.5975	0.20500	0.021	0.0664	1.1287
Pix2pix vs. Ref	0.0553	0.24671	0.996	−0.5838	0.6945
vs. Without pre-reconstruction processing	−0.2854	0.20143	0.490	−0.8073	0.2365
vs. MSBF	−0.5975	0.20500	0.021	−1.1287	−0.0664
Source of Variation	df *	Sums of Squares	Mean Square	F	*p*
Processing	2	10.394	5.197	4.269	0.015
Dose	1	0.023	0.023	0.019	0.890
Processing × Dose	2	0.127	0.063	0.052	0.949
Error	199	242.237	1.217	-	-

* CI: confidence interval, dependent variable: FWHM value. * df: degree of freedom, dependent variable: FWHM value.

**Table 2 diagnostics-12-00495-t002:** Spatial resolution of microcalcifications (MCs) performances of tomosynthesis processing methods. (Tukey–Kramer test; *p* < 0.05 indicates a significant difference).

MCs: 0.23 mm					
Variable	Difference	Standard Error	*p*	95% CI *	
				Lower Limit	Upper Limit
Ref vs. Without pre-reconstruction processing	0.0363	0.16756	0.996	−0.3972	0.4697
vs. MSBF	−0.6841	0.16835	0.000	−1.1197	−0.2486
vs. pix2pix	−0.1218	0.16756	0.886	−0.5553	0.3116
Without pre-reconstruction processing vs. Ref	−0.0363	0.16756	0.996	−0.4697	0.3972
vs. MSBF	−0.7204	0.13778	0.000	−1.0768	−0.3640
vs. pix2pix	−0.1581	0.13681	0.656	−0.5120	0.1959
MSBF vs. Ref	0.6841	0.16835	0.000	0.2486	1.1197
vs. Without pre-reconstruction processing	0.7204	0.13778	0.000	0.3640	1.0768
vs. pix2pix	0.5623	0.13778	0.000	0.2059	0.9188
Pix2pix vs. Ref	0.1218	0.16756	0.886	−0.3116	0.5553
vs. Without pre-reconstruction processing	0.1581	0.13681	0.656	−0.1959	0.5120
vs. MSBF	−0.5623	0.13778	0.000	−0.9188	−0.2059
Source of Variation	df *	Sums of Squares	Mean Square	F	*p*
Processing	2	20.275	10.138	15.045	0.000
Dose	1	0.275	0.275	0.409	0.523
Processing × Dose	2	0.642	0.321	0.477	0.621
Error	243	163.734	0.674	-	-

* CI: confidence interval, dependent variable: FWHM value. * df: degree of freedom, dependent variable: FWHM value.

**Table 3 diagnostics-12-00495-t003:** Spatial resolution of microcalcifications (MCs) performances of tomosynthesis processing methods. (Tukey–Kramer test; *p* < 0.05 indicates a significant difference).

MCs: 0.29 mm					
Variable	Difference	Standard Error	*p*	95% CI *	
				Lower Limit	Upper Limit
Ref vs. Without pre-reconstruction processing	0.0238	0.11462	0.997	−0.2728	0.3203
vs. MSBF	−0.8083	0.11462	0.000	−1.1048	−0.5118
vs. pix2pix	−0.1958	0.11462	0.321	−0.4923	0.1007
Without pre-reconstruction processing vs. Ref	−0.0238	0.11462	0.997	−0.3203	0.2728
vs. MSBF	−0.8321	0.09359	0.000	−1.0742	−0.5900
vs. pix2pix	−0.2196	0.09359	0.091	−0.4617	0.0225
MSBF vs. Ref	0.8083	0.11462	0.000	0.5118	1.1048
vs. Without pre-reconstruction processing	0.8321	0.09359	0.000	0.5900	1.0742
vs. pix2pix	0.6125	0.09359	0.000	0.3704	0.8546
Pix2pix vs. Ref	0.1958	0.11462	0.321	−0.1007	0.4923
vs. Without pre-reconstruction processing	0.2196	0.09359	0.091	−0.0225	0.4617
vs. MSBF	−0.6125	0.09359	0.000	−0.8546	−0.3704
Source of Variation	df *	Sums of Squares	Mean Square	F	*p*
Processing	2	26.778	13.389	42.460	0.000
Dose	1	0.092	0.092	0.291	0.590
Processing × Dose	2	0.028	0.014	0.044	0.957
Error	245	77.256	0.315	-	-

* CI: confidence interval, dependent variable: FWHM value. * df: degree of freedom, dependent variable: FWHM value.

## Data Availability

All relevant data are within the manuscript and its nonpublished material files.

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
