# Peer review of "Evaluation of a Generative Adversarial Network to Improve Image Quality and Reduce Radiation-Dose during Digital Breast Tomosynthesis"

_diagnostics, 2022, doi:10.3390/diagnostics12020495_

Round 1
Reviewer 1 Report
This paper builds upon an emerging body of literature where deep-learning methods are being applied to improve breast tomosynthesis image quality with one objective being to lower the radiation dose of the exam since it is often used as a screening modality. This is a preliminary study that uses phantoms alone, imaged on a single vendor system. This makes the results only narrowly applicable, with limited potential for translation to the clinic without substantial additional work. However, there appear to be some contributions beyond other literature as the authors use real images that are referenced to clinically relevant imaging techniques. Although the authors need to clarify the specific advantages of pix2pix, the use of a widely available cGAN as the basis for the algorithm appears to be an advantage for widespread investigation and development of similar algorithms in the field. Several opportunities exist to improve upon or extend this work, but this study could stand as a proof-of-principle after appropriate revision. Substantial revision to wording is required in some sections to improve grammar and for clarity. Other specific comments follow.
Abstract
Line 14: the first sentence of the abstract isn’t a sentence. Please revise. Grammar errors and inaccurate uses of wording/terms are found throughout. A careful review is suggested, especially of the Abstract and Discussion section.
Line 15: “pix2pix pre-reconstruction processing was compared with conventional filtered back projection (FBP)…” Please revise wording for accuracy as pix2pix pre-reconstruction processing wasn’t compared to FBP.
Since AEC is a commonly known acronym, please revise usage. I think it will be confusing for readers to equate AEC with the reference dose value. Perhaps define a new acronym, such as AECrd = (Automatic Exposure Control reference dose) to refer to a dose value.
Line 23 “ FWHM images…” the FWHM is a metric and not an image, please reword for clarity
“the minimum in the deterioration” – what is this?
The last sentence of the abstract could be removed. It is somewhat redundant, but also requires revision for accuracy of use of language and to appropriately reflect the results (see comments on the Conclusion below).
Introduction
Check reference numbering as it appears it may not match with the current list. E.g., line 49-50: Wu et al isn’t ref [9].
Can you put pix2pix in context vs other cGAN approaches? Does it have any particular technical advantages? Or are the main benefits related to its wide availability and relatively easy implementation?
Line 77: I would argue that this is an overstatement: “further decreases in patient doses and improvements in detection rely on a new algorithm that improves image quality by incorporating suitable processing.” Please reword as there are other approaches to dose reduction (e.g., new detector types, alternative x-ray sources, etc.) under development that could also lower dose. Image processing is just one approach, albeit a practical one, that could optimize the use of existing DBT technology.
Methods
Pg 5, line 189 Does “straightforward on the detector” refer to the “0 degree” projection? As in, the central projection view of the 15 view set? Did you experiment with testing the MSE and SSIM of other projection views? The performance at the extreme angles will depend on the object shape, so it may be relevant to check the sensitivity of optimization according to the choice of the projection view angle with respect to the detector.
Line 208: Please describe some details about the FBP algorithm. Is this a custom script, or a Matlab package, etc.?
Similarly, what is the source of the MSBF algorithm? Please provide some details.
The CNR background ROI placement could influence the CNR absolute value substantially given the variable background in the BR3D phantom. For more comparable results between lesion sizes and phantom thicknesses it is recommended that either multiple background regions, or an annulus around the lesion, be used, rather than a single background ROI.
How many phantom images were used for training/testing the algorithm? Were the same images used for both training and evaluation?
Results
Figure 1. The section title “3.2 Image quality” appears at the end of the caption.
Figure 3. The 4.7 mm lesion image in the bottom left row (with_pix2pix_pre-reconstruction_processing_h) seems to be missing, and the image of the 3.9 mm lesion appears to have been repeated twice.
Figures 2 to 4. It seems redundant and unnecessary to include the following text in the caption, as it could be stated in the Methods section, or as text within the main manuscript in the Results section, “The X-ray source was moved along the image vertically. In the displayed areas, the full width at half-maximum (FWHM), contrast-to-noise ratio (CNR), peak signal-to-noise ratio (PSNR), and structural similarity (SSIM) were determined (PSNR and SSIM analyzed the area of the entire image.)”
Figures 2 to 4. What do you mean by “display variety”? Are you referring to the image contrast, or grayscale window/level?
Figures 5 to 7 – label the y axis to specify that it’s the FWHM, such as “FWHM (# pixels)”
Figures 8 to 10 – for bar charts, please denote any values with statistically significant differences (e.g., such as adding * above the bar)
Discussion
Pg 20. Line 583: Does “observational study” refer to a clinical study using human subjects?
Pg. 21, line 585: The sentence “we plan to investigate the correlation between physical evaluation (spatial resolution and contrast)” seems to be incomplete. The correlation between physical evaluation and what? Or do you mean to assess the correlation between the physical parameters? Such as spatial resolution vs contrast?
Line 586: The sentence “We believe that pix2pix pre-reconstruction processing will optimize the acquisition protocol in future DBT imaging” requires some re-wording for accuracy. The processing won’t do the optimization directly, which is how it sounds now. So you could revise to something like, We believe that pix2pix pre-reconstruction processing will allow for optimization of the use of dose in future DBT imaging.
The paragraph from lines 563 to 567 doesn’t seem to belong in the Discussion. Perhaps in the Introduction.
Line 568 to 572: Please clarify what you mean by “largest normal contribution” and “original contribution” as these terms haven’t been introduced earlier in the paper. I see that this text is largely borrowed from Gomi, J. Biomed Sci and Eng, 2014. I would suggest that an explanation of the cause of artefacts in FBP isn’t necessary in this paper. The authors could consider simply suggesting that the use of an IR algorithm be considered in future work to reduce image artefacts, and possibly to further reduce noise.
The sentences about the limitation of the phantom materials are largely redundant (and also largely adapted from Gomi, J. Biomed Sci and Eng, 2014). This limitation could be shortened to one sentence saying something about how the BR3D materials may not accurately represent breast tissue x-ray attenuation or breast tissue structure, so studies using real patient images are required.
It would be interesting if you could postulate on if/how the results may translate to clinical images. The phantom images provide a very limited set of characteristics. How might the algorithm training/optimization differ when using patient images?
Please expand the scope of the limitations. Additional limitations:
-single vendor system evaluated
-analysis of MSE and SSIM for the central projection only
-evaluation of the in-focus slice of the reconstruction volume only, such that any out-of-plane features/artefacts are not considered
-use of FBP algorithm only, without optimization. This leaves opportunities for future work.
-the evaluation of mass lesions is quite limited. Evaluation using a wider variety of sizes, shapes and margin types (e.g., smooth, spiculated) is recommended. In addition, the CNR metric isn’t ideal to capture the influence of lower frequency signal and noise/structure. The authors mention that the structural noise component may be reflected in the background noise measurement. However, this is only at the scale of the ROI. Ideally, one would use more advanced methods such as a detectability index (e.g., see ref #26 by Gao et al, 2021), where the influence of anatomical noise can be included. In addition, the definition of the CNR does not account for the human perception of signals and noise. This would be better represented by a detectability index, where the spatial frequency content of the signal can also be included, as well as estimates for the response of the human visual system to image signal and noise across the spatial frequency spectrum.
Pg 21, line 589: Braudi et al? Do you mean Barufaldi et al?
Also, the referenced paper used Laplacian fractional entropy as a metric to evaluate the simulation of anatomical noise according to a Perlin noise algorithm.
What do you mean by “anatomic noise technology”? Anatomical noise isn’t a technology.
Overall, the discussion about anatomical noise is not directly connected with the work here. Please expand your discussion to clarify your intent.
Pg 21, line 594”: “investigating the simultaneous reduction of GAN…” I assume GAN is a typo here. Please revise this sentence for accuracy.
Conclusion
The authors overreach the conclusions made in the final sentence, “…can be integrated into the clinical application workflow to accelerate image processing and reduce noise while maintaining excellent image quality in radiologic imaging of the breast.”
First, by the use of the term “accelerate” do you mean to suggest that the image processing is faster? Because I didn’t find this to be clear from the study.
Second, “radiologic imaging of the breast” suggests that this could be broadly applied to any radiological breast imaging, but only DBT was investigated here.
Third, the absolute image quality level wasn’t evaluated here. All evaluations were relative. So while one would hope that the AEC would provide excellent image quality, this wasn’t strictly evaluated.
But overall, my main concern is that the sentence suggests that the denoising algorithm could be directly translated to the clinic. The results don’t support this given the use of phantom images. The wording should be modified to state something like, that “pix2pix shows promise for integration into the clinical application workflow to reduce image noise while maintaining image quality in breast tomosynthesis.”
Author Response
We wish to express our appreciation to the reviewer for offering insightful comments, which have helped us improve the paper significantly. Please see the attached file for the details of the response.

Reviewer 2 Report
Dear authors,
Your work is really amazing, but this manuscript has too many self citations.
There are similarities with "Gomi T, Sakai R, Hara H, Watanabe Y, Mizukami S. Usefulness of a Metal Artifact Reduction Algorithm in Digital Tomosynthesis Using a Combination of Hybrid Generative Adversarial Networks. Diagnostics (Basel). 2021;11(9):1629. Published 2021 Sep 6. doi:10.3390/diagnostics11091629"
Author Response

(The authors gave the same response as above.)

Round 2
Reviewer 1 Report
The revision to this paper has adequately addressed my comments and I do find the paper to be improved overall. The point-by-point response to the comments is appreciated.
The corrections to spelling and grammar have improved the text, however, there were some additional errors introduced within the new revisions. My comments focus on a correction of these new errors, so there are only minor changes suggested.
The proposed revisions are outlined below:
Line 15/16: “In this study, we evaluated the improvement of image quality in digital breast tomosynthesis under low-radiation dose[s] conditions…”
Line 81: I didn’t intend for the example list to be exhaustive, so this should be qualified, e.g., “Therefore, further decreases in patient doses and improvements in detection rely on [innovations such as] a new detector type, …”
Line 99/100: What is DT in this sentence? I didn’t find the acronym introduced here. “Noise and radiation-dose reductions using deep learning for DT of the breast and metal artifact reduction are possible”
Line 221/222: The newly inserted “all” should be moved to later in the sentence, “We used MATLAB (MathWorks, Natick, MA, USA) to reconstruct and process [all] images…”
Line 610: The wording has become awkward for this sentence and needs revision. My suggestion: “Second, we did not perform a clinical study using human subjects.”
Figure 2 to 4 captions: This sentence requires further modification for clarity, “The display referring to the image contrast of the BR3D phantom was changed for visual comparison of the contrast and background gray levels.”
There is confusion caused by choice of wording here. Currently, “contrast” is used with two different meanings. The first refers to the measure of a difference between image intensities, while the second usage refers to the ‘signal’ (i.e., calc or mass) in the image to be detected. Please revise accordingly.
Author Response
We wish to express our appreciation to the reviewer for offering insightful comment, which have helped us improve the revised paper significantly.
Please refer to the attached file for details.
